# Fostering Sustainability and Critical Thinking through Debate—A Case Study

**Alfonso Rodriguez-Dono** [1,2,*] and **Antoni Hernández-Fernández** [3]

1   Department of Civil and Environmental Engineering, Universitat Politècnica de Catalunya, 08034 Barcelona, Spain

2   Institute of Environmental Assessment and Water Research (IDAEA), Spanish National Research Council (CSIC), 08034 Barcelona, Spain

3   Institut de Ciències de l'Educació, Universitat Politècnica de Catalunya, Av. Doctor Marañón 44-50, 08028 Barcelona, Spain; antonio.hernandez@upc.edu

*   Correspondence: alfonso.rodriguez@upc.edu

**Abstract:** Transversal competences such as sustainability or critical thinking have become more important in the last decades in University teaching. The objective of this article is to assess the effectiveness of debate as a teaching method capable of fostering such competences in engineering students. To do this, a debate activity has been held facing two reasonable positions: Sustainable Development versus Degrowth. The research methodology consisted of performing this activity in the classroom (with 13 students in this particular case study), and conducting some tests before and after the debate that served as feedback to assess the effectiveness of the debate on the learning process. This feedback is composed of different documents, including a pre-post test, a specific survey about the debate, a pre-post knowledge survey and the Student Evaluation of Educational Quality (SEEQ) survey. A methodology for the organization of the debate is proposed and the answers of the students to these feedback documents are analyzed. As it can be inferred from the different surveys, the debate has been an efficient learning tool to acquire knowledge and to develop sustainability and critical thinking competences.

**Keywords:** sustainable development goals; critical thinking; debating; higher education; flipped classroom; jigsaw

## 1. Introduction

Transversal competences became more important in the last decades in education policy documents [1,2], including higher education [3]. These transversal competences include, among others, critical thinking, sustainability and teamwork [4]. How to develop these transversal competences and, at the same time, comply with the specific competences of each subject has been a recurrent objective in higher education [3,5,6].

Moreover, critical thinking is increasingly viewed as one of the most important aspects of higher education [7,8], believed to play a central role in logical thinking, decision-making, and problem solving [9,10]. Hence, it is crucial for scientists, engineers and, in fact, citizens in general to acquire a certain critical capacity that helps them know how to distinguish between evidence and opinion, and identify whether the conclusions are supported or not by the evidence [11]. Many indicators show the difficulty that people have in mastering this type of thinking [12], as is often evident in social and political debates. Without the ability to think critically, people may fall prey to many exaggerated, unfounded and dubious claims [13]. Furthermore, critical thinking is considered one of the most important skills for college graduates to become effective contributors in the global workforce [14]. Therefore, the promotion of critical thinking has a transversal significance in education [7].

On the other hand, there is evidence in favor of debate as an effective method to develop transversal competences—mainly critical thinking—of students in higher educa-

tion. For example, some authors present debate as a valuable teaching–learning strategy for teaching critical thinking and improving communication skills [15,16]. Some of this evidence is related to nursing [17], occupational therapy [18], sociology [19,20], athletic training education [21], dental education [22,23], marketing [24] or accounting [25], but it is a method rarely used in engineering subjects, a fact that this paper intends to mitigate. In addition, another study concluded that gender did not have a significant effect on the students' critical thinking skills developed with debate [26].

Another transversal competence to foster is 'sustainability and social commitment', in which the goals are to understand the complexity of the economic and social phenomena typical of the welfare society, be able to relate well-being to globalization and sustainability, and acquire skills to use technology, economics and sustainability in a balanced and compatible way [6]. In fact, engineering subjects usually contain many issues that pose future challenges for humanity in the following decades, regarding their sustainability. Some examples of these issues can be the depletion of medium-term fossil fuels, the climate changing, recycling, etc. These are transversal and quite complex problems with repercussions in many fields, and are not sufficiently known socially, so their inclusion in the engineering education programs is totally justified [27]. In short, it is a paradigm shift affecting the entire political, economic and social framework. However, despite the declaration of good intentions and policy developments at the national, regional and international level, little has been achieved in terms of embedding education for sustainable development holistically in the curriculum [28].

Thus, the objective of this article is to assess the effectiveness of debate as a teaching method capable of fostering the competences of critical thinking and sustainability in engineering students. The methodology developed following this purpose is intended to be carried out in master's in engineering courses, but can be adapted to any course. These engineering courses have typically around 15 students per year, which is the approximate number of students taken into account for this work. Nevertheless, the number of students involved in the particular academic year of this study has finally been 13 students.

In the following, the methodology followed during the course, including the design of the debate and its evaluation, as well as the collection of a number of feedback documents, is explained (Section 2). Then, the feedback documents are analyzed and discussed, in terms of both educational significance and scientific basis (Section 3).

## 2. Methodology

Engineering subjects usually contain many issues that pose future challenges for humanity in the following decades regarding their sustainability. In previous courses, many of these elements for reflection were raised in classes. In addition, classical activities (for example, reading an article) were usually set, requesting the personal opinion of the students, with some subsequent class discussion on these issues.

However, an improvement in the structure and quality of these discussions should be made to improve the acquisition of critical thinking skills and sustainable development philosophy. This improvement could be achieved by conducting a formal debate [29,30]. In fact, debate is an active learning technique that encourages critical thinking, because it consistently demands the questioning, examining and restructuring of knowledge [31], and the debater learns not to trust assertions, realizing that problems have more than one side [32]. With this purpose, in this article, students prepare a debate in which two sides are formed to defend different opinions, and a third group analyzes critically their arguments.

All of this effort could be seen as a reduction of the short time that professors have for transmitting the specific knowledge of their subject. However, moving from declarative knowledge (i.e., professors giving a speech) to procedural knowledge, where learning is achieved through thinking and debating, the students acquire a deeper learning, which is associated with high quality learning [8].

On the other hand, although it has undoubtedly brought wellbeing for many people—perhaps not all—the problem of economic growth in recent decades is that it is associated

with an increase in energy consumption, a consumption of natural resources and a production of waste. All of the latter, in addition to already causing various tangible problems, has led to the belief that conventional economic growth models were not sustainable and, therefore, could not be maintained for a long time. This is how the idea of Sustainable Development emerges, understood as an economic growth that is sustainable over time [33].

Nowadays, the idea of economic growth is still maintained in goal number 8 of the Sustainable Development Goals [34]. In order to make this economic growth sustainable, the concept of 'decoupling' is proposed (Figure 1). Decoupling means using less resources per unit of economic output, and reducing the environmental impact of any resources that are used, or economic activities that are undertaken [35]. This is the recipe of the main international organizations and countries that, based on technology, aim to mitigate the collateral damage of economic growth, through energy savings and efficiency, recycling, etc.

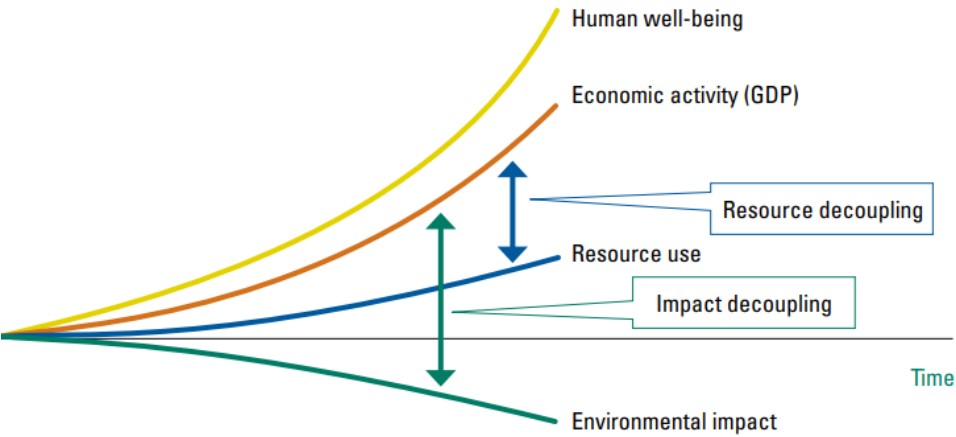

**Figure 1.** The 'decoupling' concept [35].

However, there are critics of this global vision, who consider it unrealistic [36–47]. These critics argue that economic growth cannot be sustainable. Thus, they introduce the concept of Degrowth (or at least a steady-state, zero-growth economy) to ensure the sustainability of the system.

### 2.1. Proposal

The methodology referred to here is within the context of the so-called flipped classroom [48–52]. A class discussion is held facing two reasonable positions. For the first course, the aim is to face the idea of Sustainable Development (SD) versus that of Degrowth (D). In the following, a teaching methodology is proposed to foster the competences of critical thinking and sustainability in engineering subjects. We have addressed sustainability in a broad perspective, although we have focused more on environmental aspects.

During the weeks leading up to the debate, several exercises are carried out to prepare it, using a jigsaw (or puzzle) strategy [53,54], which basically consists of the students receiving information and having to synthesize it, to be able to explain it later to their classmates. The jigsaw cooperative model shows significant positive changes in learning outcomes and student learning styles compared to conventional learning models [55–57].

In this way, three teams of students have been formed (Sustainable Development, Degrowth and Critical Thinking–CT) that have received material regarding their field of knowledge for 4 weeks. The students received the materials one week ahead of the class date. Then, every week, during the class, the students discussed the material of the week during approximately one hour, according to the scheme shown in Table 1.

**Table 1.** Weekly discussions scheme.

| Section | Time (min) | Description |
|---|---|---|
| Team meeting | 20 | Discussion among members of each team (e.g., SD team) on the basic concepts contained in their material and preparation to explain them to the other teams. |
| Group meeting | 20 | Different groups form with at least one student from each team (ideally three students per group, in case all the teams have the same number of students). In turns, they explain each other the basic concepts related to their studied materials and they discuss it. |
| Test | 20 | Each team answers a test including 6 basic questions related with the studied material (2 questions related to SD; 2 to D and another 2 to CT) |

### 2.2. Design of the Debate Session

The methodology of the debate is inspired by the work of Hall [58]. Students are divided into three teams of approximately 5 members each; in this particular case study, the first team had 5 members and the other teams 4 members each, from a total of 13 students. The first group defend the option of Sustainable Development (SD), the second the Degrowth (D) and the third group are the Moderators (or CT team). Each of the teams are told to make a presentation (in PowerPoint, for example) of 15 min, exposing their thesis.

After the initial presentations setting the initial arguments, a second part starts which includes the argument shifts. Finally, the debate ends with a critical analysis of the arguments presented [30,58]. The debate has been recorded so the arguments can be further analyzed.

#### 2.2.1. Moderators

The moderators are responsible for introducing the debate, moderating it during its development and carrying out a critical analysis of the arguments put forward.

The proposed structure of the debate session, inspired in the work of Hall [58], can be seen in Table 2. The group of moderators start the session, introducing the topic to be discussed and making a presentation (of 15 min) about what is critical thinking (CT), what is and what is not an argument and what fallacies we should avoid [59,60]. After this presentation, the other two groups make their own presentations defending their thesis.

**Table 2.** Structure of the debate.

| Discourse | Time (min) | Team | Description |
|---|---|---|---|
| Introduction | 15 | Moderators | Debate presentation and basic aspects of CT |
| SD defense | 15 | SD team | Statement of arguments in defense of SD |
| D defense | 15 | D team | Statement of arguments in defense of D |
| Preparation | 15 | All | Preparation of the second part of the debate |
| Synthesis | 5 | Moderators | Synthesis of the arguments of each team |
| SD defense | 5 | SD team | Counter-argument to the D team |
| D defense | 5 | D team | Counter-argument to the SD team |
| Preparation | 5 | SD and D teams | Preparation of the last shifts of debate |
| Free shifts | 10 | SD and D teams | Last shifts of counter-argument |
| Preparation | 15 | Moderators | Preparation of the final analysis |
| Final analysis | 5 | Moderators | Critical analysis of the arguments |
| Final shift | 5 | All | Extra shifts to qualify (optional) |

After this first part, the teams and the moderators have some time to prepare the second part of the debate, which begins, by the moderators, with a summary of the

arguments presented by the two other teams, and then proceed to give the word to each team to present their counter-arguments.

During the debate, the moderators are responsible for their development, having to control the times and give the floor to one or another discussion team, as appropriate (see Table 2). As for the control of the times, it is advisable to have some flexibility (for example, 5 min or so) depending on how the debate is developing, while trying to ensure that the total time does not exceed 2 h, which is the estimated duration of the debate.

The moderators may ask open questions to the teams if deemed necessary with the aim of energizing the debate or unblocking some point of disagreement.

At the end of the debate, the moderators have 15 min to prepare a critical analysis of the arguments put forward by each team, justifiably pointing out those that have been more solid and convincing. Finally, they will give the floor to the teams in case they have any nuances to add.

### 2.2.2. Discussion Teams

After the presentation of the moderators, the other two groups will carry out their presentations, defending their thesis. Each presentation lasts 15 min.

The objective of each discussion team should be to persuade the audience—in this case the other group and the moderators—about their point of view. To do this, in addition to carefully examining the existing literature and preparing their own arguments, they should prepare to counter the arguments of the opposing team and, if possible, refute them.

Thus, each team makes an initial presentation and then tries to counter-argue the main ideas of the opposing group in the second part of the debate, always respecting the shifts assigned to them by the moderators.

It must be borne in mind that the position of each individual in the debate does not have to coincide with his or her personal position. Moreover, with the purpose of developing critical thinking, it is often desirable that it does not coincide, because it is about debating critically and not falling into the so-called confirmation bias [61]. Thus, each group must find good arguments to defend the position entrusted to it.

### 2.3. Academic Evaluation

In order to assess the performance of each student in the debate session, individual performance is taken into account, so all students must participate in the debate. All teams are told to deliver their respective presentations before the debate. In addition, moderators are told to deliver a short document with a critical analysis of the arguments presented during the debate, either at the end of the debate or a few days later, agreeing with the teachers the exact date.

Regarding the evaluation of the performance of the students during the debate, the authors have developed a couple of evaluation rubrics (Appendix A) in which a few ideas from the evaluation rubric used by Hall [58] were taking into account. Note that we decided to evaluate the competence of communication, because we think it is an important competence in a debate, although this is not related to the main aim of this article.

For the evaluation of the debaters, the moderators must fill in an evaluation rubric (Figure A1) for each student, and this should be consistent with their final critical analysis of the arguments. For the evaluation of the moderators, on the other hand, the debate teams must cover another evaluation rubric (Figure A2) for each student who participates in the moderation of the debate. In the case of moderators, their ability to analyze critically the arguments provided by the debaters is considered.

The evaluation rubrics are divided in three different sections, i.e., communication, argument and knowledge. In general, we could state that these three sections correspond to the competences of communication, critical thinking and sustainability, respectively, although in the case of the moderators, knowledge refers to their knowledge about what critical thinking is and how a proper argument should be made. Nevertheless, even if these rubrics evaluate the performance of the students during the debate in terms

of communication, argument and knowledge, the acquired knowledge of the students, including the moderators, about sustainability is more objectively assessed with the pre-post test (Section 2.4).

*2.4. Feedback*

Besides the aforementioned documentation, some additional documents were designed to try to assess the effectiveness of the debate on the learning process, including a pre-post test, a specific survey about the debate, a pre-post knowledge survey and the Student Evaluation of Educational Quality (SEEQ) survey [62]. These documents are shown in Appendix B.

1.  Debate survey: a specific survey with personal opinions of the students about the debate has also been performed after the debate. The questions included in this non-anonymous survey are shown in Figure A4. The results of this survey can help to make a qualitative assessment on the perceptions of the students regarding their learning process and the competences acquired;
2.  Pre-post knowledge survey (Figure A5): the objective of this anonymous survey is to assess how familiar the student is with some basic concepts related to sustainability and, thus, how this plays a role in the debate. This survey has been taken before the debate (at the beginning of the semester) and after the debate (post-survey). Note that when we say, 'before the debate', we are including the preparation weeks prior to the debate itself, in which some activities, including class discussions, were developed. With this survey we can have a measure of the perceived learning of the students after the debate compared with their perceived knowledge before the debate;
3.  Pre-post test (Figure A3): a non-anonymous pre-post test study has been carried out to assess the effectiveness of the debate in the learning process. This test consisted of delivering 20 questions to the students before and after the debate, and comparing the results. In fact, to assess the influence of the discussion groups during the classes and not only the influence of the final debate, the pre-test has been made at the beginning of the semester. As it can be seen, it includes Likert-type questions and open questions. This test can serve as a measure of what the students have learnt during the debate and their critical analysis capacity, which can be induced combining their answers with the arguments given during the debate. This induction is performed by the authors as they try to analyze the responses of the students, taking into account their arguments;
4.  SEEQ survey: this well-known anonymous survey has also been taken by the students at the end of the semester. The Students' Evaluation of Educational Quality (SEEQ) survey is a tool for collecting students' evaluations of college/university teaching quality, measuring nine distinct components of teaching effectiveness.

## 3. Analysis and Discussion

### 3.1. Debate Survey

The first part of the debate survey is about the preparation of the debate. According to the results of this survey (Figure A4), the students studied in depth the documentation provided by the professor (an average of 4 points out of 5), although every team put specific emphasis in the documentation regarding their role in the debate. Moreover, the students used an average of 8 extra documents to broaden the documentation given by the professor.

Regarding the comments of the students about the debate, after some organization of the comments by categories, we have been able to synthesize the comments in a word cloud (Figure 2) and a summary table (Table 3).

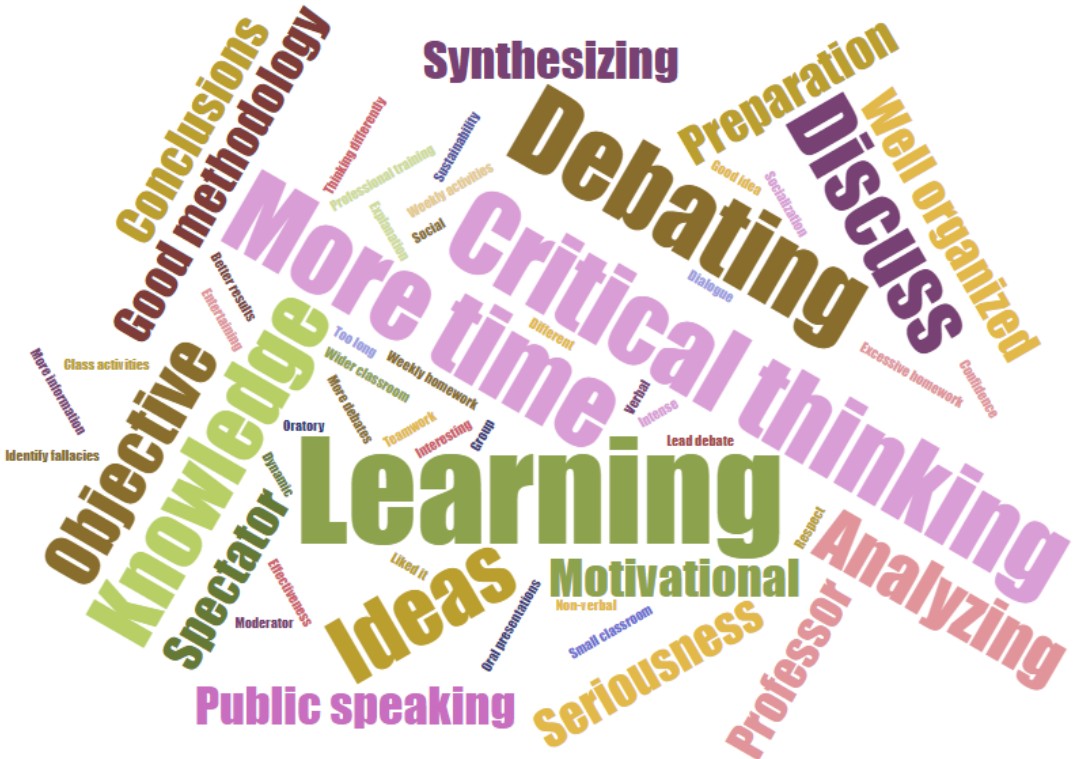

**Figure 2.** Word cloud corresponding to the debate comments from the students.

**Table 3.** Student Comments on Debate.

| | |
|---|---|
| General perception | • An efficient learning tool to acquire knowledge (5)<br>• Learning in more depth than doing an exam (4)<br>• Motivational to learn and prepared well for the debate (3)<br>• Good for learning how to debate<br>• Helps to think critically and be more social (4)<br>• Professional training, since daily work is more a debate than an exam<br>• Interesting and entertaining (2)<br>• There should be more debates in other subjects (3)<br>• Debate should constitute a higher part of the evaluation<br>• Did not seem a good idea at first, but liked it at the end<br>• Good idea to have weekly activities to prepare the debate<br>• It was very well organized |
| Liked best | • Learning how to debate and how to organize a debate (2)<br>• Learning new concepts and acquiring knowledge (4)<br>• Learning from other students and hearing their analysis of the situation<br>• Respect and seriousness in the debate<br>• Being able to acknowledge mistakes in arguments from other debaters<br>• Observing verbal and non-verbal expressions from other students<br>• Debating with other students in an intense and objective dialogue<br>• Debating important topics such as sustainability<br>• Doing something different from other subjects<br>• Weekly homework and class activities discussing what was learned (2)<br>• More dynamic classes and with better results than conventional ones<br>• Methodology and debate guidelines<br>• Generating own ideas is better than memorizing<br>• Liked that the professor was a spectator in the debate |

**Table 3.** *Cont.*

| | |
|---|---|
| Liked least | • Preliminary presentations; some of them not respecting allowed time (4)<br>• Not enough time to debate; presentations part should be another day (4)<br>• No time enough to respond questions proposed by moderators (3)<br>• Too much homework in short time (2)<br>• Defend something that you don't believe<br>• Not enough information to prepare the debate<br>• Professor should participate and lead the debate (3)<br>• Room not big enough<br>• The debate was too long |
| Developed skills compared to a conventional class | • Overall, learning; debate requires more preparation than an exam (4)<br>• Learning much more and easier than in a conventional class (2)<br>• Ability to speak in public and face discussions with confidence (3)<br>• Helps oratory and improving oral presentations (4)<br>• Learning to organize an argument, discuss it and draw conclusions (2)<br>• Learning to argue and counter-argue and to identify fallacies (2)<br>• Helps to think differently and understand people with different ideas<br>• Helps to think critically and defend a non-personal position (6)<br>• Being able to analyze and synthesize complex ideas<br>• Teamwork (2) |

Note: the numbers at the end of each sentence represent repetitions of the comment.

Regarding the general perception of the students, we can observe that most students reported explicitly that the debate was motivational and had been an efficient learning tool to acquire knowledge, compared to an exam, although they said that debate requires more preparation than an exam. However, some thought daily work is more similar to a debate, so it is better for professional training. In addition, they fed-back in favor of the weekly activities to prepare the debate (explained in Section 2.1).

Furthermore, in general they seemed to agree with the methodology and debate organization, although some of them found that there was not enough time to debate (total time for debate was planned to be 2 h). In fact, they suggested the possibility of moving the preliminary presentations part to another day, so they could have more time to debate as well as more time to meditate after hearing the exposition from the teams. Some of them also expressed that the professor should participate and lead the debate, although other student preferred that the professor was just a spectator and that they had to be able to organize the debate themselves, having been given the general guidelines. There were additional comments, somehow less representative, as can be seen in Figure 2 and Table 3.

Concerning the skills developed, the students thought that they learned much more in a debate than in a conventional class or doing an exam. In fact, they reported their disposition to have debates in more subjects, and that the debate should have a higher weight in the overall evaluation (in this subject it was 20%). They showed much more confidence in debating that they had before, and considered themselves more prepared to debate. In addition, they considered that it helped to develop critical thinking, oratory, teamwork, empathy, public speaking, debating skills, etc.

*3.2. Pre-Post Knowledge Survey*

In Figure 3, we can observe the mean values corresponding to the perceived knowledge of the 13 students that participated in the debate. In blue we can see the mean values before the debate for the 20 different concepts included in the survey (Figure A5), and in green color the mean values after the debate.

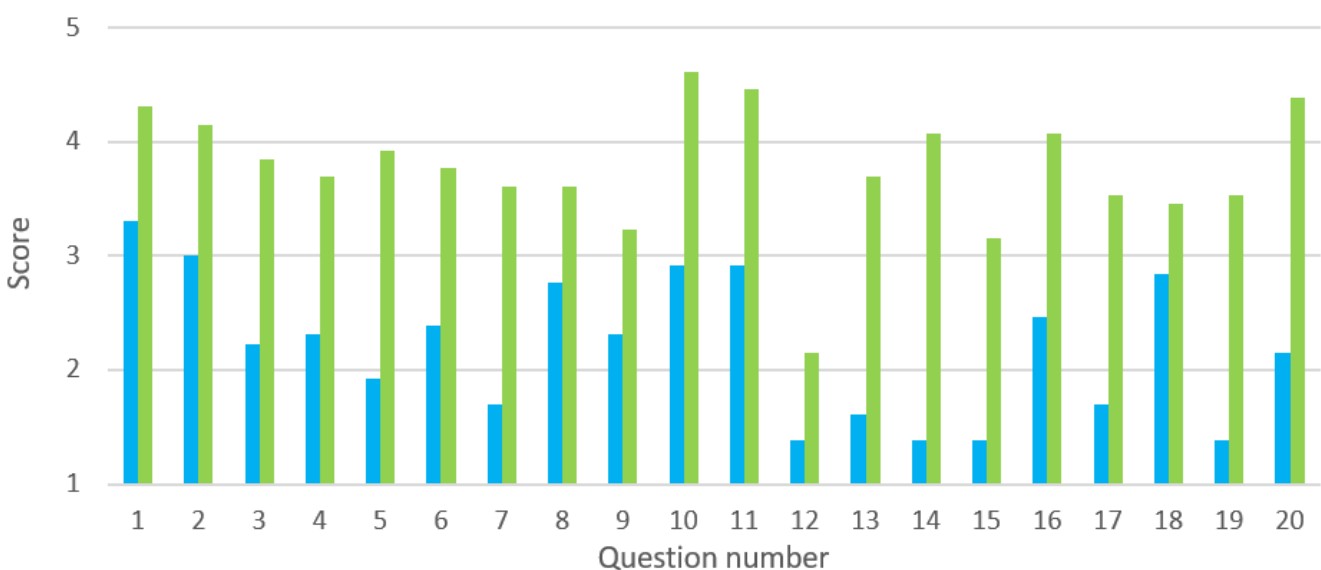

**Figure 3.** Mean values for the knowledge survey, before (blue) and after (green) debate.

Overall, the students had a mean perceived knowledge of 2.1 (from 1 to 5) before the debate, at the start of the semester. After the debate, the overall value grows until 3.7. We cannot compare this to a conventional class because we had not made this survey in previous years, but at least it is clear that the perceived knowledge grows overall and in every concept, and especially in concept 14 (environmental externalities), with a growth of 2.7 points.

On the other hand, concept 18 (landscape integration) had the least growth in mean perceived knowledge (0.7), although was one of the most known concepts in the pre-survey, together with Sustainable Development Goals (10), Gross Domestic Product (11) and mine restoration (8). However, Gini coefficient (12), environmental externalities (14), strip-mining (15) and Jevons paradox (19) were the least known concepts in the pre-survey.

Furthermore, the most well-known concepts (over 4.4 out of 5 in mean perceived knowledge) in the post-survey after the debate were planned obsolescence (20), Sustainable Development Goals (10) and Gross Domestic Product (11). Moreover, Gini coefficient (12) was the least well-known by the students. However, regarding Gini coefficient, we observed that 3 students reported having substantial or vast knowledge on the concept, while 6 students reported no knowledge at all. The explanation for this may be that only some students from the SD and D teams learned the concept while preparing for the debate, while, e.g., the moderators (CT team) and some other students, did not learn it since this concept did not appear explicitly in the debate.

Finally, 12 of these 20 concepts (1,2,3,5,6,9,10,11,14,16,17,20) appeared explicitly in the debate, while concept 7 (EROEI) appeared but not explicitly. The rest of the concepts of this survey appeared during the activities for the debate preparation, but did not appear in the debate itself. Interestingly, we observed that concept 12 (Gini coefficient), analyzed above, did not appear during the debate. In addition, some other concepts not included in the survey appeared in the debate, such us 'cradle to cradle', 'technological optimism' or the 'new green deal'.

Overall, the responses of the students to this survey have shown that, in average, their perceived knowledge has been increased by 77%.

### 3.3. Pre-Post Test

3.3.1. Likert-Type Questions

In Figure 4 we can observe the scores (from 1—strongly disagree, to 5—strongly agree) related to the first Likert-type 10 statements (Figure A3). The overall results including all students are presented, as well as the results corresponding to each team.

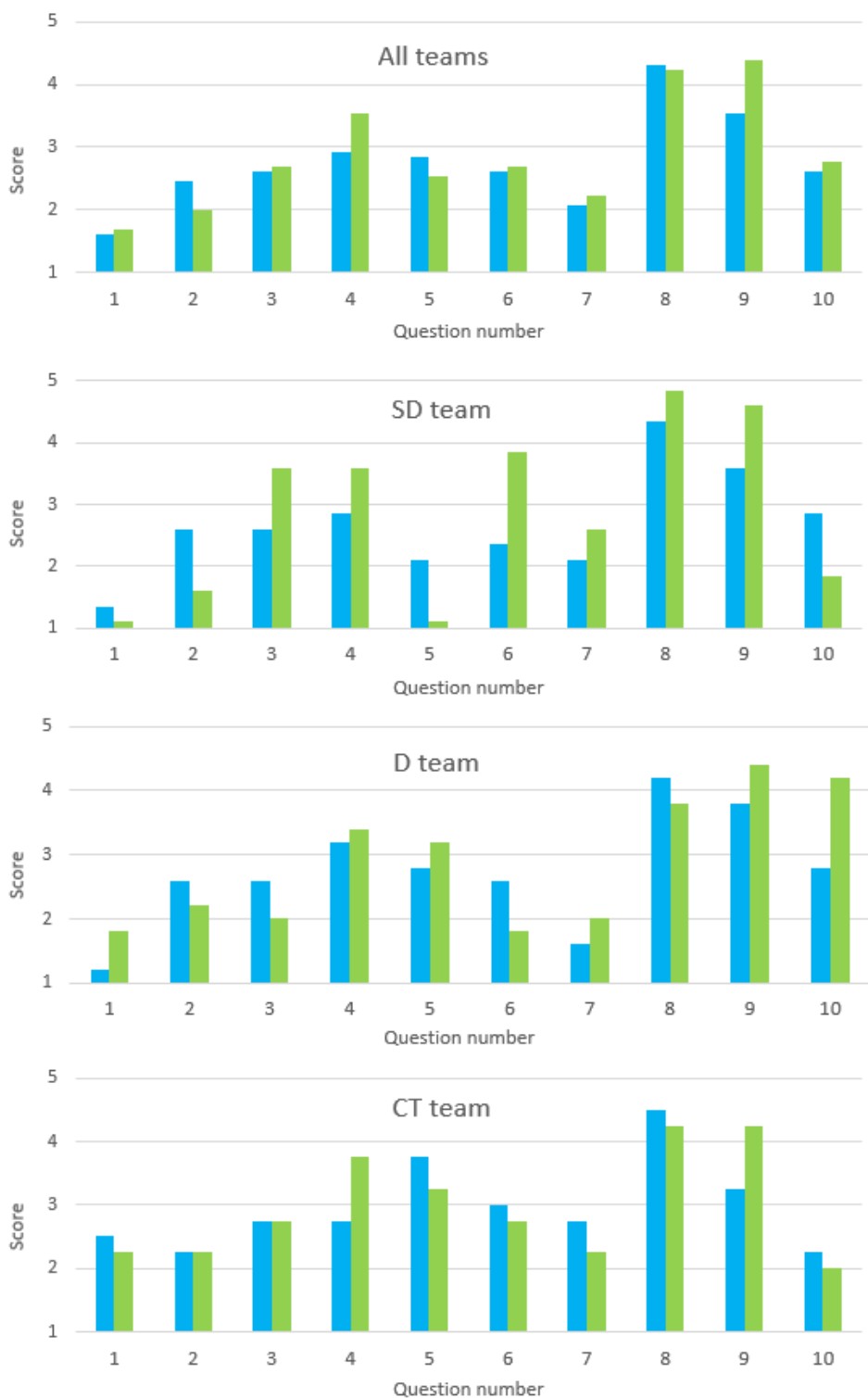

**Figure 4.** Mean scores corresponding to the pre-post test, before (blue) and after (green) debate. Scores are given for all students (above) and for each team of students (SD, D and CT). Scores range from 1—strongly disagree to 5—strongly agree.

While in the previous section we have seen how the students' perception of having learned improved (Figure 3), we can observe that in the pre-post test, critical thinking is promoted by inducing the students to position themselves in the face of different statements. The answer to some of these statements may be clear in the scientific literature (such as question 1, for example) but in other cases can be more controversial (such as question 4,

in which nuclear fission technology is mentioned, or questions 7 and 10 mentioned below). Thus, students are induced to assess different arguments for and against different statements regarding sustainability. Promoting the development of critical thinking implies precisely avoiding ideological dogmatism and being able to change your mind in the face of solid arguments that may be raised.

Perhaps the two main questions in the debate are question 7 and question 10. Can it be sustainable an economic growth based on increased energy consumption? In general, the students seem to find it unlikely. However, to the question of whether a zero-growth, or even degrowth, policy is needed, students parted from a seemingly open perspective at the beginning of the course; after debate, SD team students did not think it is necessary, meanwhile D team students thought it is necessary.

According to Allen et al. [63], global warming is correlated with the production of greenhouse effect gases due to the burning of fossil fuels. In general, the students were aware of this fact at the beginning of the course, as can be observed in the responses to question 1 (global warming has no correlation with the production of greenhouse effect gases due to the burning of fossil fuels) in Figure 4.

Fossil fuel reserve depletion times for oil, gas and coal are approximately 35, 37 and 107 years, respectively [64], and a peak of conventional oil production before 2030 appears likely [65]. Moreover, according to the International Energy Agency's Sustainable Development Scenario, natural gas production should decrease by 2040, even if the share of shale gas, obtained by fracking, is expected to grow in the total global gas production [66]. Students seemed to be already aware to some extent of these facts at the beginning of the course, as can be observed in the responses to question 2 (Figure 4). However, students of the SD team seemed to have a clearer mind about the subject after the debate, probably because they have further developed thinking about how to grow in a sustainable way, in order to defend their position.

Regarding question 3 (Figure 4), technology can help to solve the problem of fossil fuel depletion and their related gas emissions, although this technological aid will be mainly focused on energy efficiency and renewable energy production (Figure A7). The students did not show a clear opinion, neither before nor after the debate. However, there is also the question on how much we can grow in a planet with limited resources [67]. This may explain why students from the D team became somewhat more pessimistic about technology, since they defend an economic degrowth, or zero growth, meanwhile the students of the SD team defend a sustainable growth, so they likely became more technologically optimistic when learning about possible solutions. On the other hand, students of the CT team seemed to be more convinced by the defense of the SD team.

Moreover, regarding nuclear fission energy, uranium reserves are not sufficient to guarantee the supply for more than thirty years [68] and the issue of radioactive waste disposal is still not resolved [69,70]. As for nuclear fusion technology, it is at the level of research and far from being available at the operational level [71]. However, the IEA's Sustainable Development Scenario sees nuclear energy play an important role in decarbonizing the power sector, in tandem with renewables [66]. Students are somewhat more convinced about nuclear energy's paper in the future energy supply after the debate, although they still show doubts (question number 4 in Figure 4).

As mentioned above, renewable energies are key for Sustainable Development. However, it is true that they may not be technologically mature enough yet to cover the whole global energy demand, especially the (yet) non-electrical demand, which is expected to be about 70% by 2040, even in the IEA's Sustainable Development Scenario [66]. Following the debate, my interpretation of the results of question 5 is explained by the fact that, although students know about the importance of renewables (see the response to question number 8 in Figure 4), some are not convinced that it would be possible to substitute completely fossil fuels. That is the case of the students supporting degrowth. Nevertheless, SD team students based their defense in a 100% renewable energy future that, thanks to the advance in technology, would let economies continue growing.

On the other hand, carbon capture, utilization and storage (CCUS) have also been considered in the IEA's Sustainable Development Scenario (Figure A7). Then again, it is expected to lower up to 9% of the total $CO_2$ emissions under that scenario, so it could be considered to be merely part of the solution to global warming. Once again, different teams have different perspectives about this (question number 6 in Figure 4), since the SD team became more convinced of this solution and the D team less convinced. The CT team remained neither convinced nor unconvinced.

Concerning question 9 (Figure 4), the students became more aware of the importance of circular economy to achieve sustainable development, both in an economic growth or degrowth scenario, by changes in the way that materials are used and produced, via more efficient, lower carbon process routes [66]. This acquired awareness, together with the arguments presented during the debate, suggest a deep learning of the students of multidisciplinary complex concepts related to sustainability.

### 3.3.2. Open Questions

Here, the student responses to the open questions in the pre-post test (questions from 11 to 20) are referenced and analyzed.

Fossil fuel consumption is widely recognized as unsustainable [72]. In fact, it is estimated that it took nature about 5 million years to produce the amount of fossil fuels that nowadays we consume in just one year [73]. The median value of the answers of the students for this question is correct, so they seemed to be aware of this at the beginning of the course (question number 11; Figure 5), but responded with more precision after the debate, as shown by the reduction in the normalized standard deviation value ($-0.27$).

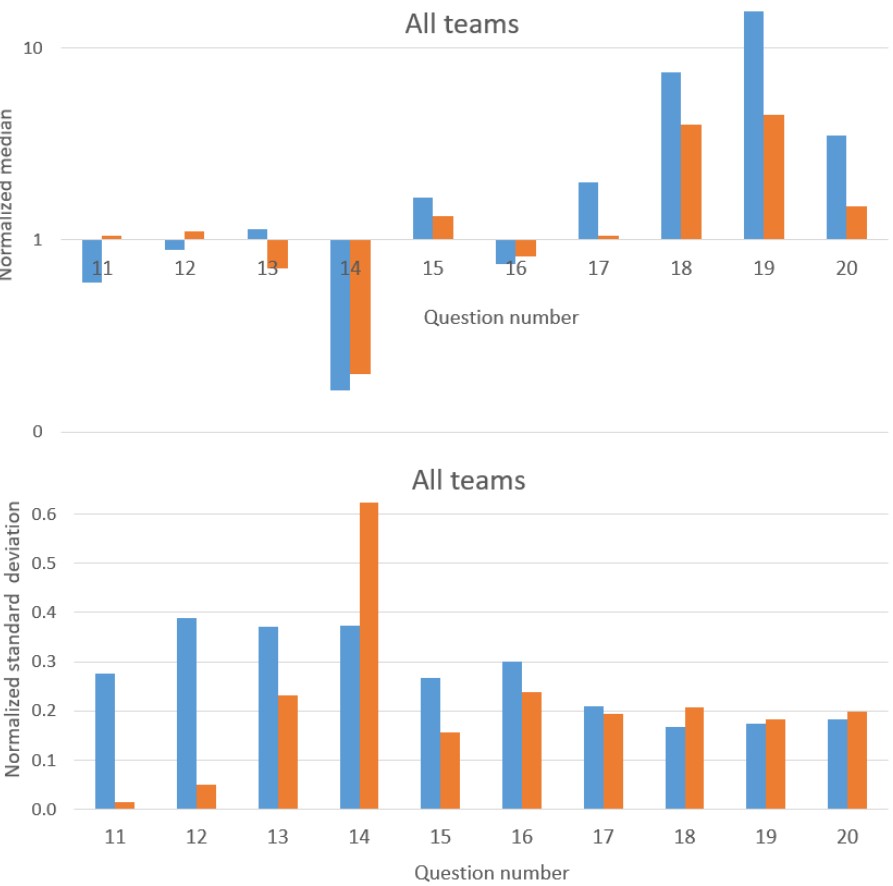

**Figure 5.** Normalized median and normalized standard deviation of the answers to the pre-post test, before (blue) and after (orange) debate. Since values are normalized, the correct answer for every question is 1.

The world's richest 1% of people own around 45% of the world's wealth [74]. Students were in general aware of this before and after the debate, but responded with more precision (−0.34 in normalized standard deviation) after the debate (question number 12; Figure 5).

It is estimated that 35% of women worldwide have experienced either physical and/or sexual intimate partner violence or sexual violence by a non-partner (not including sexual harassment) at some point in their lives [75]. Students were in general aware of this before, responding a bit more precisely (−0.14 in normalized standard deviation) after the debate (question number 13; Figure 5).

Arable land loss is estimated at 30 to 35 times the historical rate [76]. The answer of the students to this question remained far from correct after the debate (question number 14; Figure 5), as can be seen in the value of the median, although it is a bit closer to the correct answer (0.04 units closer to 1). However, the dispersion of the answers increased in this case (+0.25), which means that the students were not proven to acquire a deep learning regarding this particular concept. Nevertheless, although they have shown not to be aware of the magnitude of the problem (30–35 times the historical rate), they showed that they are aware of the problem of arable land loss, with a median response of 2 times the historical rate (even one student responded 10 times the historical rate). In addition, these values are in part affected by the responses of the moderator's team, with a median response of 1.4 times the historical rate, lower than the rest of the students.

Should the global population reach 9.6 billion by 2050, the equivalent of almost three planets could be required to provide the natural resources needed to sustain current lifestyles [77]. Students improved a little bit their response both in terms of normalized median (0.34 units closer to 1) and in terms of normalized standard deviation (−0.11, i.e., closer to zero) after the debate (question number 15; Figure 5).

About two-thirds of the world population experience severe water scarcity during at least one month of the year [78]. Students improved in part their response both in terms of normalized median (0.08 units closer to 1) and in terms of normalized standard deviation (−0.06, i.e., closer to zero) after the debate (question number 16; Figure 5).

There is not a clear influence of the debate on the responses to these questions. Most of the answers have already been surprisingly accurate (at least its median) at the beginning of the course.

Finally, there are some questions about the perception of the students on the status of countries on different indexes measuring their development (Figure A6 in Appendix C). First, the most traditional indicator: the Gross Domestic Product (GDP), followed by the GDP at purchasing power parity (PPP) per capita [79], which also takes into account the population of the country and their purchasing power. Then, more holistic indicators, such as the Inequality-adjusted Human Development Index (IHDI) [80]; and the Global Happiness Score [81]. It can be observed that the answer to these questions improved in terms of its standard deviation (questions 17–20; Figure 5).

Overall, the responses of the students to these open questions have become more precise after the debate. In fact, if all those responses are taken as a whole, their average median has been 53% closer to the correct answer and their average standard deviation has been 23% smaller.

### 3.4. SEEQ Survey

The distribution of results of the SEEQ survey [62] is shown by categories in Figure 6. It can be observed that the student responses are quite satisfactory overall, with most of the students agreeing or strongly agreeing in all categories. In fact, the average value of any of the categories is 4 or greater (out of 5). The most satisfactory category is 'group interaction', over 4.5 in average.

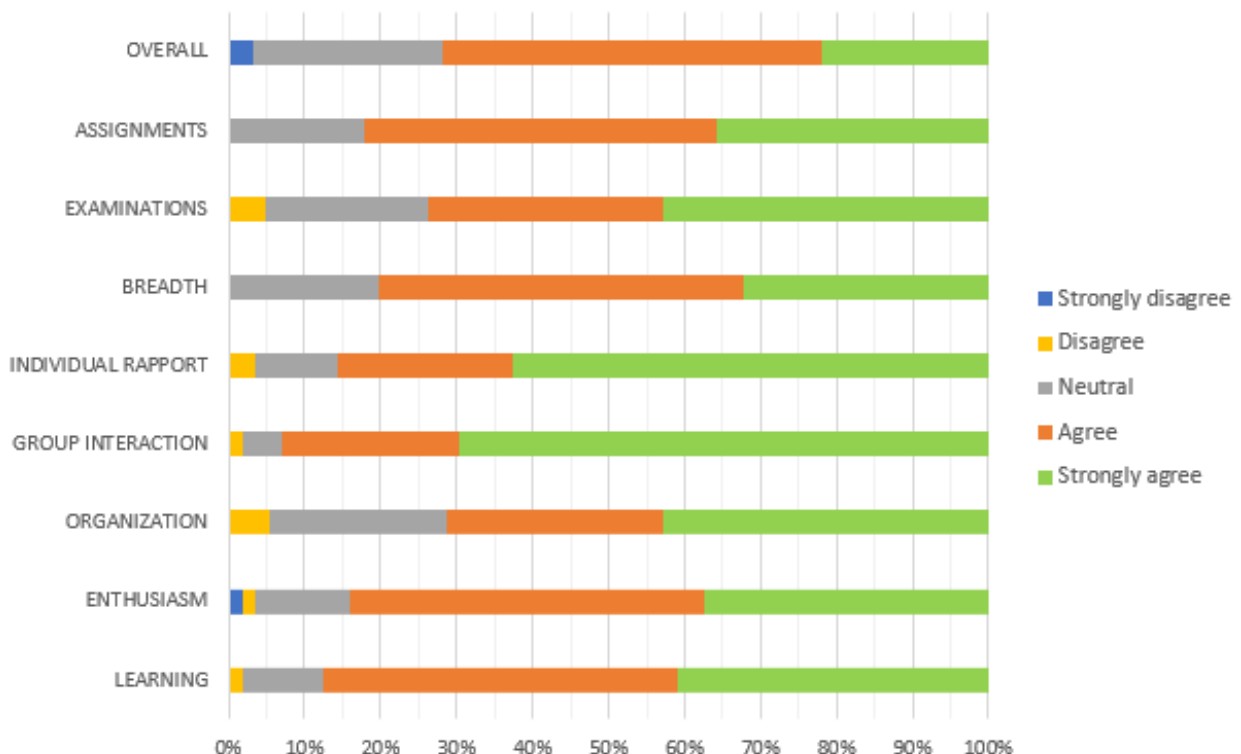

**Figure 6.** SEEQ survey results distribution by categories.

### 3.5. Academic Evaluation

The students obtained more than 85%, in average, for the three different sections evaluated, i.e., communication, argumentation and knowledge (Table 4). The D team obtained the lowest average scores in the different categories, but were still higher than 74%. In fact, making a deeper analysis of the data, the main reason for the relatively low score of the D team in the argument category has been that it was considered that they did not always provide enough evidence or examples to support their arguments.

**Table 4.** Academic evaluation average scores of students per team.

| Team | Communication | Argument | Knowledge | Total |
|---|---|---|---|---|
| D team | 79% | 74% | 83% | 78% |
| SD team | 93% | 88% | 88% | 90% |
| Moderators | 94% | 91% | 94% | 93% |
| All | 89% | 85% | 89% | 88% |

### 4. Conclusions

The objective of this article is to assess the effectiveness of debate as a teaching method capable of fostering sustainability and critical thinking competences in engineering students. To do this, a class discussion has been held facing two reasonable positions: Sustainable Development versus Degrowth. Although the sample of this case-study is somewhat limited—13 students participated in the debate—some interesting conclusions have been obtained from the field study.

The question about whether this method is more effective than other traditional methods is not addressed directly, since the results have not been compared with those of a conventional class. This can be thought of as another limitation of this work. However, if we want to know if the proposed method is effective, we can clearly say that the results of this case study add to the evidence in favor of debate as an effective method to develop

transversal competences of students in higher education [3], such as sustainability and critical thinking. In fact, as discussed below, Figure 3 (knowledge survey) shows the increased perception of students that they have learned with the debate, while the pre-post test (Figure 4) shows that, beyond searching 'the correct answer', students argue and critically reflect on sustainability.

The answers of the students to a number of questionnaires have been analyzed. As it can be inferred from the students comments (Table 3), their overall satisfaction with the debate is high. In fact, regarding the comments of the students about the debate, most students think that it has been an efficient learning tool to acquire knowledge. They also think that a debate is better for professional training. They expressed satisfaction with the methodology and debate organization, although most of them found that there was not enough time to debate. In addition, they consider that it helped to develop critical thinking, oratory skills, teamwork, empathy, public speaking, debating skills, etc. Moreover, they think that they learn much more in a debate than in a conventional class or by doing an exam.

In fact, the feelings of the students go in the direction of the investigations that empirically show that reasoning and the generation of knowledge are improved in an interactive group [82], where of course debate has a fundamental role [30], especially to promote an essential scientific attitude [83]. This is consistent with the direct observations of the authors during the debate and with the evaluation results, since the students obtained more than 85%, in average, for the three different sections evaluated, i.e., communication, argument and knowledge.

The pre-post knowledge survey (Section 3.2) and the pre-post test (Section 3.3) have shown quantitatively that students have indeed learned and, although some questions are currently in open discussion [46], they demonstrated an acquired knowledge and analysis of the situation. For example, the responses of the students to the pre-post knowledge survey have shown that, in average, their perceived knowledge has been increased by 77%. Moreover, the responses of the students to the open questions of the pre-post test have become more precise after the debate, with their average median 53% closer to the correct answer, and their average standard deviation 23% smaller. Finally, the SEEQ survey shows that students are quite pleased with the course in general.

In summary, analysis of the feedback documents have shown that students have not only learnt, but have also developed sustainability and critical thinking competences, among other transversal competences such as communication and teamwork.

**Author Contributions:** Conceptualization, A.R.-D. and A.H.-F.; methodology, A.R.-D. and A.H.-F.; writing—original draft preparation, A.R.-D.; writing—review and editing, A.R.-D. and A.H.-F.; supervision, A.H.-F. Both authors have read and agreed to the published version of the manuscript.

**Funding:** This research was supported by the project HEATSTORE (Ref.: PCI2018-092933), granted by the AEI (Government of Spain).

**Institutional Review Board Statement:** Not applicable.

**Informed Consent Statement:** Informed consent was obtained from all subjects involved in the study.

**Data Availability Statement:** Fundamental data is contained in Article. For additional details please contact the corresponding author.

**Conflicts of Interest:** The authors declare no conflict of interest. The funders had no role in the design of the study; in the collection, analyses, or interpretation of data; in the writing of the manuscript, or in the decision to publish the results.

## Appendix A

| | | Never | Rarely | At times | Often | Always |
|---|---|---|---|---|---|---|
| **Communication** | Establishes eye contact, uses a clear vocalization and an adequate voice volume | 2 | 4 | 6 | 8 | 10 |
| | Is respectful, does not interrupt and uses appropriate vocabulary and gestural language. | 2 | 4 | 6 | 8 | 10 |
| | Uses a structured and coherent discourse and manages well the times during presentation | 2 | 4 | 6 | 8 | 10 |
| | **Total communication:** *(30 points maximum)* | | | | | |
| **Argument** | Uses strong arguments to support ideas | 4 | 8 | 12 | 16 | 20 |
| | Is able to effectively counter-argue the arguments of the other team | 2 | 4 | 6 | 8 | 10 |
| | Provides evidence and examples to support arguments | 2 | 4 | 6 | 8 | 10 |
| | **Total argument:** *(40 points maximum)* | | | | | |
| **Knowledge** | Demonstrates good knowledge of the subject | 4 | 8 | 12 | 16 | 20 |
| | Demonstrates capacity for analysis and synthesis when necessary | 2 | 4 | 6 | 8 | 10 |
| | **Total knowledge:** *(30 points maximum)* | | | | | |
| | **TOTAL:** | | | | | |

**Figure A1.** Evaluation rubric for the assessment of the performance of the debaters.

| | | Never | Rarely | At times | Often | Always |
|---|---|---|---|---|---|---|
| **Communication** | Establishes eye contact, uses a clear vocalization and an adequate voice volume | 2 | 4 | 6 | 8 | 10 |
| | Is respectful and uses appropriate vocabulary and gestural language. | 2 | 4 | 6 | 8 | 10 |
| | Uses a structured and coherent discourse and manages well the times during presentation | 2 | 4 | 6 | 8 | 10 |
| | **Total communication:** *(30 points maximum)* | | | | | |
| **Argument** | Manages the shifts well during the debate, adapting decisions to the circumstances | 4 | 8 | 12 | 16 | 20 |
| | Synthesizes correctly the arguments of both teams and justifies their quality adequately | 4 | 8 | 12 | 16 | 20 |
| | **Total argument:** *(40 points maximum)* | | | | | |
| **Knowledge** | Demonstrates good knowledge of the subject | 4 | 8 | 12 | 16 | 20 |
| | Demonstrates capacity for analysis and synthesis when necessary | 2 | 4 | 6 | 8 | 10 |
| | **Total knowledge:** *(30 points maximum)* | | | | | |
| | **TOTAL:** | | | | | |

**Figure A2.** Evaluation rubric for the assessment of the performance of the moderators.

## Appendix B

**PRE-POST TEST**

> Please rate the following affirmations using the scale below:
>
> | Strongly disagree | Disagree | Neutral / No clue | Agree | Strongly agree |
> |:---:|:---:|:---:|:---:|:---:|
> | 1 | 2 | 3 | 4 | 5 |

01. Global warming has no correlation with the production of greenhouse effect gases due to the burning of fossil fuels    1 2 3 4 5

02. Current fossil fuel reserves will last very long and fracking has a great potential    1 2 3 4 5

03. Technology will find solutions to the problem of the depletion of fossil fuels, so we can continue with our current lifestyle    1 2 3 4 5

04. Nuclear energy −especially fusion− is the future solution for energy supply    1 2 3 4 5

05. Renewable energies are very limited, expensive and involve the use of immense areas representing a large part of the territory    1 2 3 4 5

06. Underground storage of $CO_2$ can solve the problem of global warming    1 2 3 4 5

07. Economic growth based on increased energy consumption can be sustainable    1 2 3 4 5

08. Renewable energy (wind, solar, geothermal, etc.) is the future of energy consumption    1 2 3 4 5

09. Circular economy is an important tool to achieve sustainable development    1 2 3 4 5

10. Zero-growth −or even degrowth− is necessary in order to achieve an actual sustainable development    1 2 3 4 5

Open questions:

11. Around how many years it took nature to produce the fossil fuels that nowadays are consumed in just one year, globally?

12. The world's richest 1% people own around... which percentage of the world's wealth?

13. Worldwide, around which percentage of women and girls aged below 50 have experienced physical or sexual violence?

14. Nowadays, arable land is estimated to decrease... how much faster than historical rates?

15. If the current global population growth is maintained and it reaches nearly 10000 million by 2050, how many planets Earth could be required to provide the natural resources needed to sustain current lifestyles?

16. Water scarcity affects which percentage of the global population?

17. Name the top three countries with the highest gross domestic product

18. Name the top three countries with the highest gross domestic product (at purchasing power parity) per capita

19. Name the top three countries with the highest Inequality-adjusted Human Development Index (IHDI)

20. Name the top three countries with the highest Global Happiness Score

**Figure A3.** Pre-post test.

## DEBATE SURVEY

Please rate the following affirmations using the scale below (except the numerical answers):

| Strongly disagree | Disagree | Neutral / No clue | Agree | Strongly agree |
|:---:|:---:|:---:|:---:|:---:|
| 1 | 2 | 3 | 4 | 5 |

1. Have you studied in depth the documentation provided by the professor on Sustainable Development to prepare the debate?　　1 2 3 4 5

2. Have you studied in depth the documentation provided by the professor on Degrowth to prepare the debate?　　1 2 3 4 5

3. Have you studied in depth the documentation provided by the professor on Critical Thinking to prepare the debate?　　1 2 3 4 5

4. Have you broaden the bibliography provided by the teacher to prepare the debate?　　1 2 3 4 5

5. With how many extra documents have you broaden the bibliography on Sustainable Development to prepare the debate?　　1 2 3 4 5 or more

6. With how many extra documents have you broaden the bibliography on Degrowth to prepare the debate?　　1 2 3 4 5 or more

7. With how many extra documents have you broaden the bibliography on Critical Thinking to prepare the debate?　　1 2 3 4 5 or more

|  | Personal comments about the debate |
|:---:|:---|
| Your general perception | |
| What did you like the best? | |
| What did you like the least? | |
| What skills do you think the debate helps you develop with respect to a conventional class? | |

Note: Minimum 20 words in each section.

**Figure A4.** Debate survey.

## PRE-POST KNOWLEDGE SURVEY

Please rate the following affirmations using the scale below:

| No knowledge at all | Poor knowledge | Moderate knowledge | Substantial knowledge | Vast knowledge |
|:---:|:---:|:---:|:---:|:---:|
| 1 | 2 | 3 | 4 | 5 |

1. Sustainable Development　　　　　　　　　1　2　3　4　5

2. Degrowth　　　　　　　　　　　　　　　　1　2　3　4　5

3. Critical thinking　　　　　　　　　　　　　1　2　3　4　5

4. LCA (Life Cycle Analysis)　　　　　　　　1　2　3　4　5

5. Circular economy　　　　　　　　　　　　1　2　3　4　5

6. Ecological footprint　　　　　　　　　　　1　2　3　4　5

7. EROEI (Energy Return on Energy Investment)　1　2　3　4　5

8. Mine restoration　　　　　　　　　　　　　1　2　3　4　5

9. HDI (Human Development Index)　　　　　1　2　3　4　5

10. SDG (Sustainable Development Goals)　　1　2　3　4　5

11. GDP (Gross Domestic Product)　　　　　1　2　3　4　5

12. Gini coefficient　　　　　　　　　　　　1　2　3　4　5

13. ECG (Economy for the Common Good)　1　2　3　4　5

14. Environmental externalities　　　　　　　1　2　3　4　5

15. Strip mining　　　　　　　　　　　　　1　2　3　4　5

16. Environmental assessment　　　　　　　1　2　3　4　5

17. Embodied solar energy (eMergy)　　　　1　2　3　4　5

18. Landscape integration　　　　　　　　　1　2　3　4　5

19. Jevons paradox　　　　　　　　　　　　1　2　3　4　5

20. Planned obsolescence　　　　　　　　　1　2　3　4　5

**Figure A5.** Pre-post knowledge survey.

**Appendix C**

| # | GDP | GDP (PPP) per capita | IHDI | Global Happiness Score |
|---|---|---|---|---|
| 1 | USA | Macao | Norway | Finland |
| 2 | China | Luxembourg | Switzerland | Denmark |
| 3 | Japan | Singapore | Ireland | Norway |
| 4 | Germany | Qatar | Germany | Iceland |
| 5 | UK | Ireland | Hong Kong | Netherlands |
| 6 | France | Cayman Islands | Australia | Switzerland |
| 7 | India | Switzerland | Iceland | Sweden |
| 8 | Italy | Norway | Sweden | New Zealand |
| 9 | Brazil | UAE | Singapore | Canada |
| 10 | Canada | San Marino | Netherlands | Austria |
| 11 | Russia | USA | Denmark | Australia |
| 12 | Korea, Rep. | Hong Kong | Finland | Costa Rica |
| 13 | Australia | Brunei | Canada | Israel |
| 14 | Spain | Bermuda | New Zealand | Luxembourg |
| 15 | Mexico | Iceland | UK | UK |
| 16 | Indonesia | Netherlands | USA | Ireland |
| 17 | Netherlands | Denmark | Belgium | Germany |
| 18 | Saudi Arabia | Austria | Liechtenstein | Belgium |
| 19 | Turkey | Germany | Japan | USA |
| 20 | Switzerland | Sweden | Austria | Czech Rep. |
| 21 | Poland | Belgium | Luxembourg | UAE |
| 22 | Sweden | Kuwait | Israel | Malta |
| 23 | Belgium | Australia | Korea, Rep | Mexico |
| 24 | Argentina | Canada | Slovenia | France |
| 25 | Thailand | Finland | Spain | Taiwan |
| 26 | Austria | Saudi Arabia | Czech Rep. | Chile |
| 27 | Iran | Bahrain | France | Guatemala |
| 28 | Norway | UK | Malta | Saudi Arabia |
| 29 | UAE | France | Italy | Qatar |
| 30 | Nigeria | Malta | Estonia | Spain |

**Figure A6.** Rankings of countries in different global indicators.

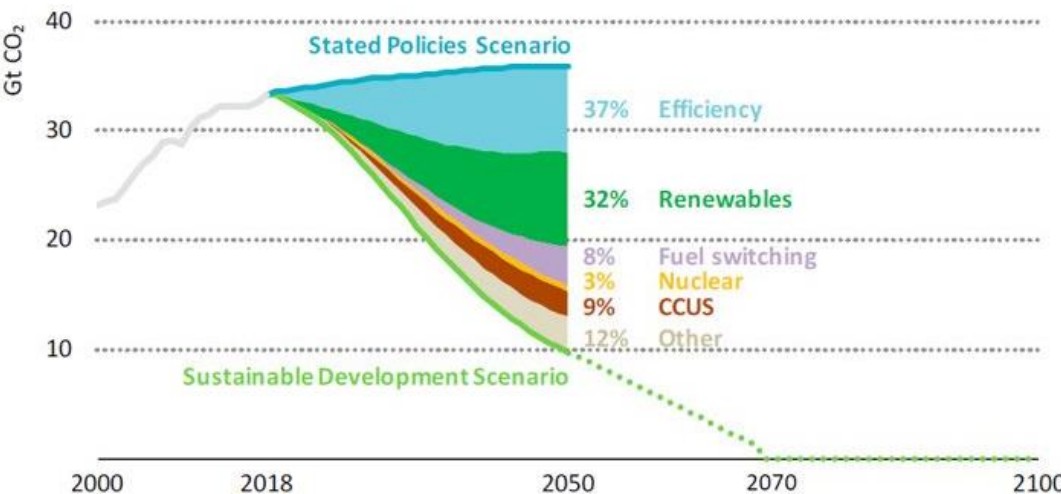

**Figure A7.** Energy-related $CO_2$ emissions under the Sustainable Development Scenario and measures needed to reduce them [66].

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
