# Peer review of "Fostering Sustainability and Critical Thinking through Debate—A Case Study"

_sustainability, doi:10.3390/su13116397_

Round 1
Reviewer 1 Report
Dear author(s),
I have read your paper with much interest the paper titled “Fostering sustainability and critical thinking through debate –a case study”. The paper aims to assess "the effectiveness of debate as a teaching method capable of fostering sustainability and critical thinking".
However, there are some issues that I noticed must be improved in order to I be able to recommend the paper for publication
Please find below major points in the article which needs clarification / refinement / reanalysis, rewrites and/or additional information and suggestions for what could be done to improve the article.
General remarks
- I strongly recommend:
- to describe (in a more detailed and concise manner) the research methodology (both in abstract and in manuscript): the research method - if it is a case study, to detail accordingly the research design (besides the way of organizing the debate, as a teaching method), participants, research instruments, data collection, data interpretation and data analysis.
- to highlight the research questions, associated with the purpose
- to refining the research purpose (“objective”) so as to be correlated with the title (or vice versa)
- to refining the conclusion section in order to highlight how the obtained results respond to the research questions and show how the debate teaching method is capable of fostering sustainability and critical thinking.
- a wider documentation on the approached topic (introduction part) highlighting results obtained in other studies and the importance of studying the topic / the novelty brought / the gap covered / the problem to which the present research responds.
- a clearer highlighting of the which sustainability competencies referred to in the performed research
- to complete the results presented with those obtained from the Academic evaluation (Figure A1, A2) and to interpret them in correlation with the other results obtained. Also, in the results part, I recommend to highlight how the results shows that the debate teaching method is capable of fostering sustainability and critical thinking
- to mention the limitations of the study
- In the section "3.3. Pre-post test" I recommend focusing the discussion on an analysis of how the results obtained show "the effectiveness of debate as a teaching method capable of fostering such competencies", rather than on the specific content of the questions. addressed to the participants in the questionnaire.
Specific remarks
Line 46-73: I recommend to move in the section “2. Methodology”
Line 83: If you agree, you can replace “the idea is to” with “the aim is to”
Line 113: I recommend to highlight the total weeks
Line 120: I recommend mentioning the number of participants in each group (al line 428 it is mentioned “Although the sample of this case-study is somewhat limited –13 students”
Line 195: I recommend to replace “Figure B1” with “Figure B2”.
Line 196: I recommend to replace “Figure B2” with “Figure B3”.
Line 202: I recommend to replace “Figure B3” with “Figure B1”.
Line 214: I recommend to add “(Figure B2)” after “According to the results of this survey,”
Line 240 (Table 3): For a better quantification of the results, I recommend highlighting the number of iterations for each comment (how many times each iteration appeared in the answers provided by the participants). This would strengthen the assessments made (line 220 – “in general”, line 232 – “most of them”, line 235 “some of them”)
Line 254: I recommend to replace “Appendix B” with “Figure B3”
Line 257” I recommend to mention the number of the obtained responses (N=?)
Line 258: I recommend to mention if the value “2.1” refers to the Mean parameter
Line 264, 270: I recommend mentioning the value for the average where assessments are made (e.g.: after “least growth” – line 264, after “most well-known” – line 270 and so on).
Line 292: I recommend to mention “Figure B1” after “Likert-type statements”
Line 359: I recommend to remove the Figure 5
At line 361, 372 - there is talk about student awareness, and at line 203 it is mentioned that this tool "assess the effectiveness of the debate in the learning process". I suggest adding an explanation/comment for readers (for an easier understanding) of how the variation of awareness shows (/”contributes to”) the "effectiveness of the debate in the learning process".
Line 392, 395: I recommend mentioning the value for the parameter (Normalized median/standard deviation) where assessments are made (e.g.: after “a little bit” – line 392, after “a little bit” – line 395 and so on).
Line 388: “the students to this question remained far from correct after the debate” - I recommend to mention how to explain this?
Line 420, 423: I recommend to keep just one of this figures Figure 7 or Figure 8)
Line 426, 431, 456: After the correlation between title and research objective (see general recommendations), I recommend to replace “transversal” with “sustainability and critical thinking”.
Line 468: Replace “Figure 1” with “Figure A1”
Author Response
Dear reviewer,
Please kindly find the response to your comments in the uploaded file.
Best regards,
Alfonso

Reviewer 2 Report
Dear Authors,
please find below my remarks regarding your article.
LINES 21-22: Key words should supplement the information contained in the title (in order to increase a "findability" of the paper) therefore, in my view, they should not be the same as words used in the title. This remark regards the first three keywords.
LINES 47-48: "These engineering courses have typically around 15 students per year, which is the approximate number of students taken into account for this work." Why not direct number? Was the study performed during one academic year or more?
LINE 116: Table 1: "Different groups form with at least one student from each team (ideally three students per group; one of each team)." - does it mean the these numbers were variable?
LINES 119-120: Again, an approximate number of students. What affected this number so it can be defined directly?
LINES 209-210: "SEEQ survey: This well-known anonymous survey..." - not so well known for a potential accidental reader sa a word or two of a explanation would be appreciated.
LINE 240: Table 3. Student Comments on Debate. I mean the lat part titles: "Developed skills respect to a conventional class" - Wouldn't it be better to use the word "compared" instead to "respect"?
LINES 251-252: There we finally have a specific sample size! "In Figure 3, we can observe the mean values corresponding to the perceived knowledge of the 13 students that participated in the debate." A bit too late to me. Should be in the section dedicated to the methods.
To conclude, I liked the article and what it brings in to teaching in general. I placed emphasis on group size issues because it should be clearly defind at the beginning, no matter if we deal with the simplest survey or highly complicated statistical analysys.
Author Response
Dear reviewer,
Please find the response to your comments in the uploaded file.
Best regards,
Alfonso

Reviewer 3 Report
Dear Authors,
The article deals with the debate methodology for the development of specific and also generic competences in a transversal way. Specifically, it deals with the development of the competences of sustainability, critical thinking and communication skills. The topic is of great current importance and the authors use current and relevant references.
The article is written in an agile, intelligible and informative style, appropriate for scientific communication.
It describes very well how the experience has been carried out and the evaluation analysis that is performed. In general, there is great coherence between the evaluation and the experience described. The discussion provides references to other research and the conclusions offer the interpretation and assessment of the results in relation to the object of evaluation.
I congratulate you on your work. Active methodologies are essential to engage students in their learning and the debate is a good example. On the other hand, the responsibility for sustainability needs to be worked on in university education.
Here are some possible improvements that I hope you will be willing to incorporate.
Possible improvements:
The objective of the paper is clearly stated in the abstract, but not in the introduction, where it is only implicitly deduced. The paper includes the study of the generic competence communication; however, the title includes sustainability and critical thinking but not communication, although it is assessed in the rubrics.
The title itself indicates that it is a case study. However, the coherence of this methodological design for the objective pursued by the article is neither mentioned nor justified later on. The discussion method is described but not the case method as such, which is the methodology used in the paper.
The importance of the development of the competences sustainability and social commitment and critical thinking for future professionals is well founded in the introduction, as well as the potential of the debate to develop them. However, the importance of the competence of communication is not justified here. On the other hand, I would add the relationship of the experience described in the paper with the development of the SDGs in engineering classrooms. The evaluation of the debate is carried out by the students themselves (self-evaluation) and among peers; however, the absence of feedback from the teaching staff is remarkable. I think this absence should be justified.
In the discussion section, there is a lack of contrast and replication of other debate experiences in the classroom, which may not have been as satisfactory.
In my opinion, figure 5 "Energy-related CO2 emissions... does not fit in the article and I would advise to remove it. From my point of view it is not directly related to the content of the article.
I understand that, in this case, sustainability is a specific competence and not just a generic-transversal one. In the rubrics that appear in the appendices, if I understand correctly, the competence of sustainability would be assessed through knowledge. The competences communication and argumentation are also assessed. It seems from the assessment indicators for argumentation that the competence critical thinking is being assessed. However, if this is the case, the term critical thinking should be changed or added, otherwise it may mislead the reader. If this is not the case, it should be added as it is part of the objectives of the paper.
Here is a summary of my proposals for improvement:
- Add in the title of the paper the generic competence communication.
- In the introduction, make the objective of the paper more explicit.
- Include the relation of the content of the article with the development of the SDGs in engineering classrooms and add it as a keyword.
- Point out the plausibility of the Case Study methodology in accordance with the objectives pursued in the paper.
- Justify the absence of feedback on the debate by the teacher.
- In the discussion, add references where the debate has not been so satisfactory and elaborate the appropriate reply.
- Remove figure 5.
- In the conclusions, point out limitations of the study and future lines of research.
- If Knowledge corresponds to Sustainability, state this in the evaluation rubric.
- Add the generic competence Critical Thinking in the evaluation rubric.
Author Response

(The authors gave the same response as above.)

Round 2
Reviewer 1 Report
Review report – Fostering sustainability and critical thinking through debate –a case study
Dear author(s),
I have read your paper with much interest the revised paper titled “Fostering sustainability and critical thinking through debate –a case study”. The paper aims to assess the effectiveness of debate as a teaching method capable of fostering sustainability and critical thinking.
The manuscript has been improved regarding the aspects addressed in the review.
However, I recommend to align the the objective highlighted at Line 61-62 (“the objective of this article is to foster the competences of critical thinking and sustainability through debate”) with the one described in the lines 13-14 and 552-553 (“The objective of this article is to assess the effectiveness (…)”.
It would be useful to:
- To be realized a wider documentation on the approached topic (introduction part) highlighting results obtained in other studies and the importance of studying the topic / the novelty brought / the gap covered / the problem to which the present research responds.
- to complete the results presented with those obtained from the Academic evaluation (Figure A1, A2) and to interpret them in correlation with the other results obtained. Also, in the results part, I recommend to highlight how the results shows that the debate teaching method is capable of fostering sustainability and critical thinking
- to focusing the discussion (In the section "3.3. Pre-post test") on an analysis of how the results obtained show "the effectiveness of debate as a teaching method capable of fostering such competencies", rather than on the specific content of the questions. addressed to the participants in the questionnaire.
